# Exploring CAR T-Cell Dynamics: Balancing Potent Cytotoxicity and Controlled Inflammation in CAR T-Cells Derived from Systemic Sclerosis and Myositis Patients

**DOI:** 10.3390/ijms26020467

**Published:** 2025-01-08

**Authors:** Janin Dingfelder, Jule Taubmann, Franziska von Heydebrand, Michael Aigner, Christina Bergmann, Johannes Knitza, Soo Park, Joseph K. Cheng, Thomas Van Blarcom, Georg Schett, Andreas Mackensen, Gloria Lutzny-Geier

**Affiliations:** 1Department of Internal Medicine 5, Hematology and Oncology, Friedrich-Alexander-Universität Erlangen-Nürnberg and Universitätsklinikum Erlangen, 91054 Erlangen, Germany; janin.dingfelder@uk-erlangen.de (J.D.); franziska.vonheydebrand@uk-erlangen.de (F.v.H.); michael.aigner@uk-erlangen.de (M.A.); andreas.mackensen@uk-erlangen.de (A.M.); 2Deutsches Zentrum Immuntherapie (DZI), Friedrich-Alexander-Universität Erlangen-Nürnberg and Universitätsklinikum Erlangen, 91054 Erlangen, Germany; jule.taubmann@uk-erlangen.de (J.T.); christina.bergmann@uk-erlangen.de (C.B.); georg.schett@uk-erlangen.de (G.S.); 3Bavarian Cancer Research Center (BZKF), 91054 Erlangen, Germany; 4Department of Internal Medicine 3—Rheumatology and Immunology, Friedrich-Alexander-Universität Erlangen-Nürnberg and Universitätsklinikum Erlangen, 91054 Erlangen, Germany; 5Institute for Digital Medicine, University Hospital Giessen-Marburg, Philipps University Marburg, 35043 Marburg, Germany; johannes.knitza@staff.uni-marburg.de; 6Kyverna Therapeutics, Emeryville, CA 94608, USA; spark@kyvernatx.com (S.P.); jcheng@kyvernatx.com (J.K.C.); tvanblarcom@kyvernatx.com (T.V.B.)

**Keywords:** chimeric antigen receptor, T-cells, cytokines, autoimmune diseases, systemic lupus erythematosus, systemic sclerosis, idiopathic inflammatory myositis

## Abstract

Systemic lupus erythematosus (SLE), systemic sclerosis (SSc), and idiopathic inflammatory myositis (IIM) are autoimmune diseases managed with long-term immunosuppressive therapies. Hu19-CD828Z, a fully human anti-CD19 chimeric antigen receptor (CAR) with a CD28 costimulatory domain, is engineered to potently deplete B-cells. In this study, we manufactured Hu19-CD828Z CAR T-cells from peripheral blood of SLE, IIM, and SSc patients and healthy donors (HDs). CAR-mediated, CD19-specific activity of these cells was evaluated in vitro by assessing cytotoxicity, cytokine release, and proliferation assays in response to autologous CD19^+^ B-cells, the CD19^+^ NALM-6 B-cell line, or a CD19^−^ U937 non-B-cell line as targets. The results demonstrated an increased proliferation of Hu19-CD828Z CAR T-cells and dose-dependent cytotoxicity against primary autologous and NALM-6 B-cells compared to non-transduced controls or co-cultures with non-B-cells. Notably, autoimmune-patient-derived CAR T-cells produced lower levels of inflammatory cytokines than healthy-donor-derived CAR T-cells in response to CD19^+^ B-cell targets. These data support the potential of Hu19-CD828Z and its therapeutic cell product KYV-101 as a therapeutic strategy to achieve deep B-cell depletion in SLE, IIM, and SSc patients, and highlights its promise for broader application in B-cell-driven autoimmune disorders.

## 1. Introduction

Chimeric antigen receptor (CAR) T-cells have revolutionized the treatment landscape for hematologic malignancies. Inspired by the remarkable success of CAR T-cell therapy in oncology, its potential applications beyond cancer are increasingly being explored [1,2,3]. Most progress is currently occurring in the field of autoimmune diseases, where existing immunosuppressive therapies are insufficient for some patients. Current therapies for autoimmune diseases often rely on long-term immunosuppression, which inadequately addresses the underlying autoimmune processes and can result in significant adverse effects.

Self-resident autoreactive B-cells play a critical role in the pathogenesis of autoimmune diseases such as systemic lupus erythematosus (SLE), idiopathic inflammatory myositis (IIM), and systemic sclerosis (SSc), leading to immune-complex-induced inflammation, organ damage, and increased mortality [4,5,6]. Historically, targeting B-cells in autoimmune diseases has primarily involved the use of monoclonal antibodies that either deplete B-cells by targeting CD20, or inhibit B-cell activation [7]. While these approaches have shown some efficacy in reducing disease activity, achieving sustained drug-free remission remains a considerable challenge, highlighting an urgent need for innovative therapeutic approaches. Additionally, while monoclonal antibodies are able to deplete B-cells in the peripheral blood, they have limited tissue penetration and are unable to deeply deplete B-cells within tissues [8]. The application of CAR T-cell technology offers a novel therapeutic avenue, with the potential for deep depletion of B-cells including autoreactive clones by targeting CD19, a pan-B-cell marker expressed across a wide range of B-cell lineage stages [9,10]. Further, CAR T-cells have been shown to drive deep-tissue-based B-cell depletion and to disrupt the architecture of tertiary lymphoid structures that may harbor pathogenic B-cells [8].

CAR T-cell manufacturing challenges have been encountered in oncology. One factor is an insufficient quantity or quality of the patient’s T-cells used to initiate the manufacturing process. This challenge can be based on the negative impact of previous immunosuppressive therapies the patient received for their disease. Hence, the effect of chronic immunosuppressants on T-cells from patients with autoimmune disease needs further investigation. In a series of 20 patients with autoimmune diseases including lupus nephritis, manufacturing of KYV-101 was robust and consistent, with a 100% success rate [11]. We have also demonstrated that anti-CD19 CAR T-cells manufactured from patients with treatment-refractory SLE are fully functional and exhibit CAR-mediated and CD19-dependent cytokine release when co-cultured with autologous primary B-cells [12,13]. Moreover, in recent case reports [14,15,16,17,18,19,20,21,22] and two small case series [23,24], CAR T-cells have been successfully used to treat SLE, IIM, SSc, rheumatoid arthritis (RA), myasthenia gravis (MG), and stiff person syndrome (SPS), demonstrating promising short-term and partially long-term outcomes up to 2 years [24].

In the present study, we investigated the production and functional activity of autologous CD19-CAR T-cells engineered with Hu19-CD828Z, a fully human second-generation anti-CD19 CAR (Kyverna Therapeutics, Emeryville, CA, USA) [25]. KYV-101, featuring the Hu19-CD828Z construct, is a CD28-based CAR T-cell product that has shown improved safety profiles in previous lymphoma studies [26] and has been applied successfully in patients with various different autoimmune diseases [22,27,28]. Hu19-CD828Z CAR T-cells were derived from the peripheral blood of SLE, IIM, and SSc patients compared to healthy subjects, aiming to provide a deeper understanding of their potential as a targeted therapeutic strategy in B-cell-driven autoimmune disorders.

Our results demonstrate that Hu19-CD828z CAR T-cells exhibit CAR-mediated, CD19-dependent activity against autologous B-cells, with comparably low inflammatory cytokine production when derived from cells from SLE, IIM, and SSc patients and healthy donors. These findings highlight the therapeutic potential of autologous, fully human CAR T-cell products, such as KYV-101, for treating B-cell-driven autoimmune disorders.

## 2. Results

### 2.1. Starting Material and B-Cell Characterization

Isolated PBMCs from peripheral blood from a total of nine patients with SLE (N = 3), IIM (N = 3), or SSc (N = 3) and three healthy donors (HDs) were characterized before T-cell enrichment (Table 1; Figure 1A). Main populations apart from T-cells included B-cells, monocytes, and natural killer (NK) cells. The proportion of T-cells and B-cells varied across individual patient groups. IIM- and SSc-patient-derived PBMCs displayed a high proportion of CD4^+^ T-cells and a low percentage of CD8^+^ T-cells (Figure 1A). Prior to co-culturing, the viability and CD19 expression of the primary B-cells were characterized. The B-cell enrichment from PBMCs was successful (≥96%), and the viability of B-cells ranged from 77% to 89% after thawing (Figure 1B). As expected, the CD19 expression level determined by MFI was higher in the CD19^+^ NALM-6 B-cell line than in primary B-cells enriched from SLE, IIM, and SSc patients or HD PBMCs. Conversely, no CD19 expression was detected in the CD19^−^ negative U937 control cell line (Figure 1C). The CD19 MFI of SLE-patient-derived B-cells was slightly higher compared to HD- (1.2-fold), IIM- (1.4-fold) and SSc- (1.6-fold) patient-derived B-cells; however, the differences were not significant (Figure 1C).

### 2.2. CAR T-Cell Manufacturing and Composition

Cryopreserved autologous CD4^+^ and CD8^+^ T-cells from patients with SLE, IIM, or SSc and HDs were transduced with a lentiviral vector to express Hu19-CD828Z, a fully human CAR directed against CD19 (Figure 2A) [25]. Both non-transduced T-cells (UNTR) and CAR T-cells derived from autoimmune patients or HDs exhibited rapid expansion until harvest on day 9 (Figure 2B). CAR T-cells derived from SSc and IIM patients demonstrated an enhanced expansion compared to those from SLE patients and HDs (Figure 2B). All T-cells were efficiently transduced with similar Hu19-CD828Z CAR expression (SLE: 62.5 ± 5.2% CAR^+^; SSc: 69.1 ± 5.1% CAR^+^; IIM: 52.2 ± 3.6% CAR^+^; HD: 54.6 ± 1.5% CAR^+^, *p* > 0.05) on day 9 (Figure 2C). While the CD4/CD8 ratio was similar for HD- and SLE-patient-derived T-cells, SSc- and IIM-patient-derived T-cells exhibited a notably lower proportion of CD8^+^ T-cells following expansion, which was already evident in PBMC starting material (Figure 1A and Figure 2D). The transduction of CAR T-cells did not alter the co-expression of the exhaustion markers TIM-3 and LAG-3 and programmed cell death protein 1 (PD-1) on day 9 (Figure 2E). The proportion of T-cells expressing these markers remained very low (<10%) for all CAR T-cells manufactured, with no differences among SLE, IIM, and SSc patients and HDs. After expansion of T-cells, the most prominent phenotype fraction was CD45RO^+^ CCR7^+^ central memory T-cells (Tcm) in both CD4^+^ and CD8^+^ CAR T-cells for all disease groups and HDs, which is consistent with previous data [12]. Compared to CAR T-cells, non-transduced T-cells from SLE and IIM patients and HDs showed a higher proportion of CD8^+^ CD45RO^−^ CCR7^+^ CD95^+^ stem cell memory T-cells (Tscm) (HD CD8^+^ UNTR: 24.4 ± 2.7% vs. HD CD8^+^ CAR: 10.1 ± 3.1%; SLE CD8^+^ UNTR: 20.7 ± 5.1% vs. SLE CD8^+^ CAR: 4.5 ± 0.6%; IIM CD8^+^ UNTR: 26.1 ± 1.8% vs. IIM CD8^+^ CAR: 11.0 ± 2.9%) (Figure 2F). The composition of memory phenotypes was similar for SLE, IIM, SSc, and HD T-cells (Figure 2F). A gating strategy for exhaustion and memory phenotyping of patient-derived CAR T-cells is demonstrated in Appendix A.

### 2.3. Proliferation

To evaluate CD19-dependent proliferation, CAR T-cells and non-transduced T-cells were co-cultured with autologous primary B-cells, the CD19^+^ NALM-6 B-cell line, or the CD19^−^ U937 non-B-cell line at an E:T ratio of 3:1 for 4 days. CAR T-cells derived from autoimmune patients or HDs demonstrated enhanced proliferation compared to the corresponding non-transduced counterparts when co-cultured with either autologous B-cells or CD19^+^ NALM-6 cells (Figure 3A,B). The proliferation level of patient-derived or HD CAR T-cells was comparable in the presence of primary autologous B-cells or NALM-6 B-cells. The respective non-transduced T-cells exhibited a similar background level of proliferation across all three co-cultures, regardless of the CD19 expression on the target T-cells. As expected, CAR T-cells exhibited a limited proliferative response comparable to non-transduced T-cells when co-cultured with the CD19^−^ U937 non-B-cell line (Figure 3C), demonstrating that the induced proliferation of Hu19-CD828Z CAR T-cells is dependent on the presence of CD19 on target T-cells.

### 2.4. Cytotoxicity

To assess the in vitro cytotoxicity of CAR T-cells derived from SLE, IIM, and SSc patients and HDs, CAR T-cells and non-transduced T-cells were co-cultured with autologous primary B-cells, CD19^+^ NALM-6 B-cells, or CD19^−^ U937 non-B-cells at E:T ratios of 1:1 and 3:1 for 16 h. All CAR T-cells exhibited CAR-mediated cytolytic activity against CD19^+^ autologous B-cells (Figure 4A) and NALM-6 B-cells (Figure 4B) at both E:T ratios. The cytolytic activity of CAR T-cells was notably higher against NALM-6 B-cells compared to autologous B-cells, indicating that the CAR-mediated cytotoxicity is dependent on CD19 expression levels (Figure 1C). In contrast, only background levels of cytolysis were observed for non-transduced T-cells (Figure 4C). As anticipated, CAR T-cells only exhibited baseline cytotoxicity when co-cultured with the CD19^−^ U937 non-B-cell line (Figure 4C).

### 2.5. Cytokine Release

To further investigate the in vitro functionality of CAR T-cells derived from SLE, IIM, and SSc patients, we analyzed cytokine secretion levels after 24 h using a multiplex cytokine assay. The cytokine release levels in the CAR T-cell co-cultures with CD19^+^ target T-cells were comparable across the autoimmune patient groups. Co-cultures of CAR T-cells with autologous B-cells revealed a pronounced increase in IFN-γ (11- to 29-fold) and TNF-α (25- to 58-fold) secretion compared to co-cultures with non-transduced T-cells. Additionally, IL-2 and IL-6 levels were significantly elevated in patient-derived CAR T-cell co-cultures. Similarly, a clear trend of increased IL-2 and IL-6 levels was observed in HD-derived CAR T-cell co-cultures compared to non-transduced T-cell co-cultures (Figure 5A). In contrast, co-culture with non-transduced T-cells led to only minimal release of TNF-α, IL-2, IL-6, and IL-1β (Figure 5A). Across all co-cultures, non-transduced T-cells showed comparable background release levels of the respective cytokines (Figure 5A–C). Upon co-culture with NALM-6 B-cells, CAR T-cells released significantly higher amounts of IFN-γ, TNF-α, IL-2, IL-6, and IL-1β compared to non-transduced T-cells (Figure 5B). Notably, CAR T-cells displayed stronger release of IFN-γ, TNF-α, and IL-2 in co-cultures with NALM-6 B-cells than with autologous primary B-cells, correlating with their higher cytolytic activity in the presence of NALM-6 B-cells and indicating a dependence on CD19 expression levels (Figure 5A,B). When co-cultured with U937 non-B-cells, CAR T-cells exhibited only low levels of cytokine release, comparable to the background levels of non-transduced T-cells (Figure 5C). Interestingly, HD CAR T-cells co-cultured with either autologous B-cells (three- to eight-fold) or NALM-6 B-cells (two-fold) tended to produce higher levels of IFN-γ than patient-derived CAR T-cells, consistent with previously published data [12] (Figure 5A,B). Notably, IFN-γ background release by HD non-transduced T-cells was slightly higher (two- to seven-fold) in NALM-6 co-culture compared to non-transduced T-cells from patients with autoimmune disease (Figure 5B).

## 3. Discussion

Anti-CD19 CAR T-cells have been shown to induce deep B-cell depletion and sustained drug-free remission in patients with systemic lupus erythematosus (SLE), idiopathic inflammatory myositis (IIM), and systemic sclerosis (SSc) [14,15,16,17,18,19,24,29]. Recently, case studies of patients with neuroinflammatory autoimmune diseases including stiff person syndrome and myasthenia gravis have also been treated successfully with the fully human, anti-CD19 CAR T-cell product, KYV-101, which contains the same CAR construct used in this study [20,22,27].

The successful generation and functional validation of Hu19-CD828Z CAR T-cells from cryopreserved autologous T-cells of patients with SLE, SSc, and IIM highlights several important findings for the advancement of CAR T-cell therapies in autoimmune diseases. Consistent with expectations from HD CAR T-cells and data from KYV-101 manufacturing showing 100% success in 20 patients with autoimmune diseases [11], T-cells derived from patients with autoimmune diseases showed robust transduction efficiency with the CAR and efficient expansion, despite variations in drug exposure, disease phenotype, and T-cell composition. Notably, T-cells from SSc and IIM patients displayed a lower CD8^+^ subset proportion compared to HD and SLE T-cells, as observed both before and after CAR T-cell expansion (Figure 1). This is consistent with the CD4/CD8 ratios of IIM and SSc patients published by Müller et al. and Volkov et al. and does not appear to have a negative impact on therapeutic outcome [18,24].

The central memory (Tcm) phenotype dominated both CD4^+^ and CD8^+^ CAR T-cell populations across all groups, which aligns with prior findings (Figure 2F) [12]. While stem cell memory T-cells (Tscm) were more prominent among non-transduced T-cells, their limited presence in CAR T-cells suggests a possible shift in phenotype following CAR transduction and expansion. While longer-term CAR T-cell persistence is associated with durable responses in oncology [30], early clinical data suggest that this duration of CAR T-cell persistence may not be needed to achieve drug-free remission in SLE patients [23]. Therefore, if the high Tcm composition in CAR T-cells from autoimmune patients indicates a favorable memory profile for therapeutic durability, further validations in ongoing patient studies are necessary to understand the impact in the autoimmune context.

Our results also reinforce the importance of CD19-dependent CAR T-cell proliferation and cytotoxicity in targeting B-cells in autoimmune patients. CAR T-cells derived from all patient groups exhibited enhanced proliferation when co-cultured with CD19-expressing autologous B-cells and NALM-6 B-cells compared to non-transduced control T-cells. Only minimal proliferation was observed in CD19-negative control co-cultures, further confirming antigen-specific activation (Figure 3). Notably, the cytolytic effect of CAR T-cells derived from patients with autoimmune disease is significantly higher than the non-transduced T-cells (Figure 4). Interestingly, CAR T-cells showed higher cytolytic activity against NALM-6 B-cells than autologous B-cells, corresponding to the differential CD19 expression levels between these target cell types. This CD19 dependency is essential in reducing off-target effects and preserving the safety of CAR T-cells in autoimmune therapy applications.

In oncology, CAR T-cells have been associated with immune-effector-cell-associated neurotoxicity syndrome (ICANS) and severe manifestations of cytokine release syndrome (CRS) [31]. Severe cases have not been observed in the initial treatment of autoimmune disease patients, likely due to the lower target burden compared to oncology [24,32]. Importantly, no adverse events, such as CRS, were documented in myasthenia gravis using KYV-101 [27]. Hu19-CD828Z, the CAR vector used in KYV-101, is an anti-CD19 CAR construct with a CD28 co-stimulatory domain, engineered to release low cytokine levels by replacing the CD28 H/TM domain with a CD8α H/TM domain. This effect was previously demonstrated in vitro and was associated with reduced serum cytokine levels in patients with B-cell lymphoma. As a result, decreased cytokine production correlated with a lower risk of CRS and reduced neurotoxicity in patients, without compromising clinical efficacy [26]. Limiting such toxicities has the potential to broaden the application of CAR T-cells in autoimmune disease and beyond.

Cytokine analysis confirmed that CAR T-cells from autoimmune patients can initiate a potent immune response upon target recognition, as evidenced by elevated levels of, for example, IFN-γ and TNF-α in co-cultures with CD19^+^ B-cells (Figure 5). Interestingly, CAR T-cells from HDs showed a higher release of INF-γ compared to CAR T-cells from autoimmune patients, suggesting possible modulation of inflammatory response due to the autoimmune disease background or prior therapies. Notably, some patients were on low-dose steroid treatment at the time of apheresis or had received steroid treatments previously, which could blunt the activity of the T-cell effector function. Despite this, the reduction efficacy in SLE patients appears to be maintained in vivo [14,23]. Additionally, the lower cytokine release in response to autologous primary B-cells compared to NALM-6 B-cells suggests a milder cytokine release profile in autoimmune-targeting CAR T-cells, which may be beneficial in reducing the risk of CRS and ICANS in clinical applications. There might also be a dependency on the higher level of CD19 expression [MFI] on NALM-6 B-cells compared to primary B-cells.

Overall, these results underscore the potential of Hu19-CD828Z CAR T-cells as a feasible and effective option for targeting CD19^+^ B-cells also in autoimmune diseases, like SLE, IIM, and SSc. The fully human autologous anti-CD19 CAR T-cell therapy, KYV-101, is currently under investigation in several multicenter clinical trials in patients with lupus nephritis, stiff person syndrome, myasthenia gravis, and multiple sclerosis to determine its therapeutic durability and safety profile.

## 4. Materials and Methods

### 4.1. Patient Material and Control Cell Lines

Three patients each with systemic lupus erythematosus (SLE), idiopathic inflammatory myositis (IIM) with antisynthetase syndrome (ASyS), and systemic sclerosis (SSc) were recruited at the Department of Internal Medicine 3—Rheumatology and Immunology, Universitätsklinikum Erlangen. Mean (±SD) age of the patients was 52.6 (±15.2) years, and mean (±SD) disease duration was 5.2 (±3.5) years. Detailed patient characteristics including organ involvement and previous and current treatments are shown in Table 1. Notably, all IIM patients share the same subtype: dermatomyositis with AsyS. In addition to cells from autoimmune patients, autologous T- and B-cells were enriched from three healthy donors’ (HDs’) peripheral blood. All procedures were performed in accordance with the Good Clinical Practice guidelines of the International Council for Harmonization and covered by license 334_18 B. All patients gave written informed consent according to CARE guidelines and in compliance with the Declaration of Helsinki. CD19^−^ U937 non-B-cells (ACC 5, DSMZ) and CD19^+^ NALM-6 B-cells (ACC 128, DSMZ, Braunschweig, Germany) served as control cell lines. They were cultivated in RPMI supplemented with 10% FBS at 37 °C in a humidified incubator with 5% CO_2_.

### 4.2. Description of Chimeric Antigen Receptor Vector

The lentiviral vector used to generate Hu19-CD828Z CAR T-cells encodes a fully human single-chain variable fragment (scFv) CD19-targeting domain, hinge and transmembrane regions from the human CD8α molecule, a CD28 cytoplasmic costimulatory domain, and a cytoplasmic CD3ζ T-cell activation domain (Figure 2A) as described previously [12,25,26]. The scFv-designated Hu19 was designed with the following sequence (5′ to 3′): human CD8α signal sequence, light chain variable region, a linker peptide (GSTSGSGKPGSGEGSTKG), heavy chain variable region. A DNA sequence encoding a CAR with the following components was designed (5′ to 3′): Hu19 scFv, part of the extracellular region and the transmembrane region of the human CD8α molecule, the cytoplasmic portion of the human CD28 molecule, and the cytoplasmic part of the human CD3ζ molecule. The coDMF5 portion of the used pRRLSIN.cPPT.MSCV.coDMF5.oPRE lentiviral vector plasmid was replaced with the Hu19-CD828Z CAR sequence.

### 4.3. CAR T-Cell Generation

Peripheral blood mononuclear cells (PBMCs) of SLE, IIM, and SSc patients and HDs were isolated from fresh blood by Ficoll. CD4+ and CD8+ T-cells were enriched from PBMCs using CD4 and CD8 microbeads (Miltenyi Biotec, Bergisch Gladbach, Germany) according to manufacturer’s instructions and cryopreserved until transduction. Then, 1 × 10^7^ CD4/CD8 T-cells were seeded in TexMACS media (Miltenyi Biotec), supplemented with IL-7, IL-15 (Mitenyi Biotec) at a concentration of 12.5 µg/L and 3% human AB serum (ccpro, Oberdorla, Germany), and activated with TransAct (Miltenyi Biotec). After 24 h, the lentivirus (Kyverna Therapeutics, Emeryville, CA, USA) was added at MOI 4. With the exception of lentivirus transduction, non-transduced control (UNTR) T-cells were treated in the same way. Cells were expanded in vitro for 9 days, cryopreserved, and thawed directly for the experiments.

### 4.4. Flow Cytometric Analyses

The following antibodies and reagents were used to characterize the immune cells: hCD3-FITC (clone SK7), hCD8-PE-Cy7 (clone SK1), hCD4-BV421 (clone RPA-T4), hCD45-APC-H7 (clone 2D1), hCD56-PE (clone MY31), hCD14-FITC (clone MφP9), 7-AAD staining solution were all purchased from BD (Becton Dickinson, Heidelberg, Germany); hCD4-APC (clone 13B8.2) was purchased from Beckman Coulter; hCD19-BV421 (clone HIB19) and hCD3-BV510 (clone UCHT1) from Biolegend (San Diego, CA, USA); and human CD19 CAR detection reagent with biotin antibody REAfinity™ PE from Miltenyi Biotec. All data were determined using the BD FACSCanto™ II Flow Cytometry System (BD Biosciences, San Jose, CA, USA) and the Kaluza Software 2.1 (Beckman Coulter, Brea, CA, USA). A gating strategy for immunophenotyping of patient-derived PBMCs is shown in Appendix A.

### 4.5. B-Cell Isolation

Autologous B-cells from SLE, IIM, and SSc patients and HDs were enriched from freshly isolated PBMCs directly after T-cell enrichment with pan-B-cell microbeads (Miltenyi Biotec, Bergisch Gladbach, Germany) according to manufacturer’s instructions.

### 4.6. Proliferation Assay

Donor-matched B-cells, CD19^+^ NALM-6 B-cells (positive control), and CD19^−^ U937 non-B-cells (negative control) were co-cultured for 4 days with patient-derived CAR T-cells at an effector:target (E:T) ratio of 3:1. Prior to seeding, CAR T-cells and non-transduced T-cells were stained with CytoPainter (CP) Cell Proliferation Staining Reagent (Abcam, ab176736, Cambridge, UK) according to manufacturer’s protocol. Flow cytometry was performed directly after seeding and on day 4. The CPDim gate was based on the UNTR T-cell monoculture on day 4 as a reference. A gating strategy illustrating the proliferation of patient-derived CAR T-cells is shown in Appendix A.

### 4.7. In Vitro Cytotoxicity Assay

Donor-matched B-cells, CD19^+^ NALM-6 B-cells, and CD19^−^ U937 non-B-cells were co-cultured for 16 h with patient-derived CAR T-cells or UNTR T-cells at an E:T ratio of 1:1 and 3:1 as technical duplicate. HD CAR T-cells served as a positive control and were treated equally. After the incubation, the cells were analyzed by flow cytometry. The percentage of live target cells was determined and used to calculate the percent cytotoxicity at each E:T ratio by the following equation: % Cytolysis = (% Live Cells (Target Alone) − % Live Cells (Sample of Interest))/% Live Cells (Target Alone) × 100. A detailed gating strategy demonstrating cytotoxicity of patient-derived CAR T-cells is presented in Appendix A.

### 4.8. Cytokine Analysis

After 24 h, the supernatants were collected from the co-culture conditions (E:T 3:1). A multiplex detection of cytokine production was performed using the MESO QuickPlex SQ 120 MM (Meso Scale Diagnostics (MSD), Rockville, MD, USA) according to manufacturer’s protocol.

### 4.9. Statistical Analysis

Data processing and graphical representation were conducted using Graphpad Prism v9. Depending on the data set, we determined statistical significance by two-tailed Mann–Whitney test, ratio paired *t*-test, Welch’s test, uncorrected Fisher’s LSD, or Tukey’s multiple comparison test. Data were considered significant if *p* ˂ 0.05.

## 5. Conclusions

In conclusion, our findings support the feasibility of generating CAR T-cells from cryopreserved T-cells of patients with autoimmune diseases, even after prior treatments with immunosuppressive therapies. Hu19-CD828Z CAR T-cells generated from SLE, IIM, and SSc patients exhibited comparably robust CD19-targeted cytotoxicity and produced low levels of inflammatory cytokines upon interaction with autologous B-cells. This profile may be well-suited for treating autoimmune patients, particularly where minimizing the risk of CRS and ICANS is essential. These results suggest that the fully human CAR construct, Hu19-CD828Z, used in KYV-101, could represent a promising therapeutic option for achieving deep disease clearance, not only for SLE but also for other autoimmune diseases, thereby expanding its potential clinical applications.

## Figures and Tables

**Figure 1 ijms-26-00467-f001:**
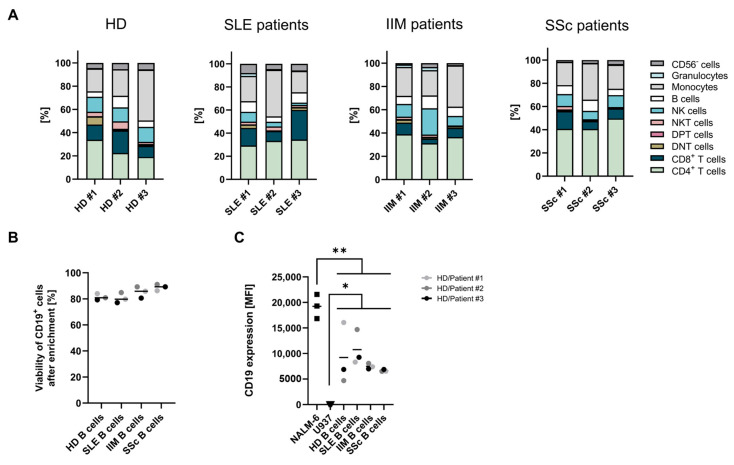
Characterization of PBMCs and autologous B-cells of SLE, SSc, and IIM patients and healthy donors. (**A**) Cellular composition of peripheral blood from systemic lupus erythematosus (SLE; N = 3), idiopathic inflammatory myositis (IIM; N = 3), and systemic sclerosis (SSc; N = 3) patients and healthy donors (HDs; N = 3) before cell enrichment. The percentages of the viable CD45^+^ population are shown, subdivided into CD4^+^, CD8^+^ T-cells, B-cells (CD19^+^), monocytes (CD14^+^), natural killer cells (NK, CD3^−^ CD56^+^), NK T-cells (NKT, CD3^+^CD56^+^), double-positive T-cells (DPT, CD4^+^/CD8^+^), double negative T-cells (DNT, CD4^−^/CD8^−^), and granulocytes (neutrophils + eosinophils; SSC^+^CD3^−^CD14^−^CD19^−^). (**B**) Viability of B-cells from SLE, IIM, and SSc patients and B-cells from HDs after enrichment. The data were analyzed by Mann–Whitney test. Bars represent median. (**C**) Flow cytometric analysis of the CD19 expression of the CD19^+^ B-cell line NALM-6, the CD19^−^ non-B-cell line U937 and autologous donor-matched B-cells. Data show individual values with indication of mean. The data were analyzed by uncorrected fisher’s LSD test, and significance is indicated by * *p* = 0.05, ** *p* = 0.01.

**Figure 2 ijms-26-00467-f002:**
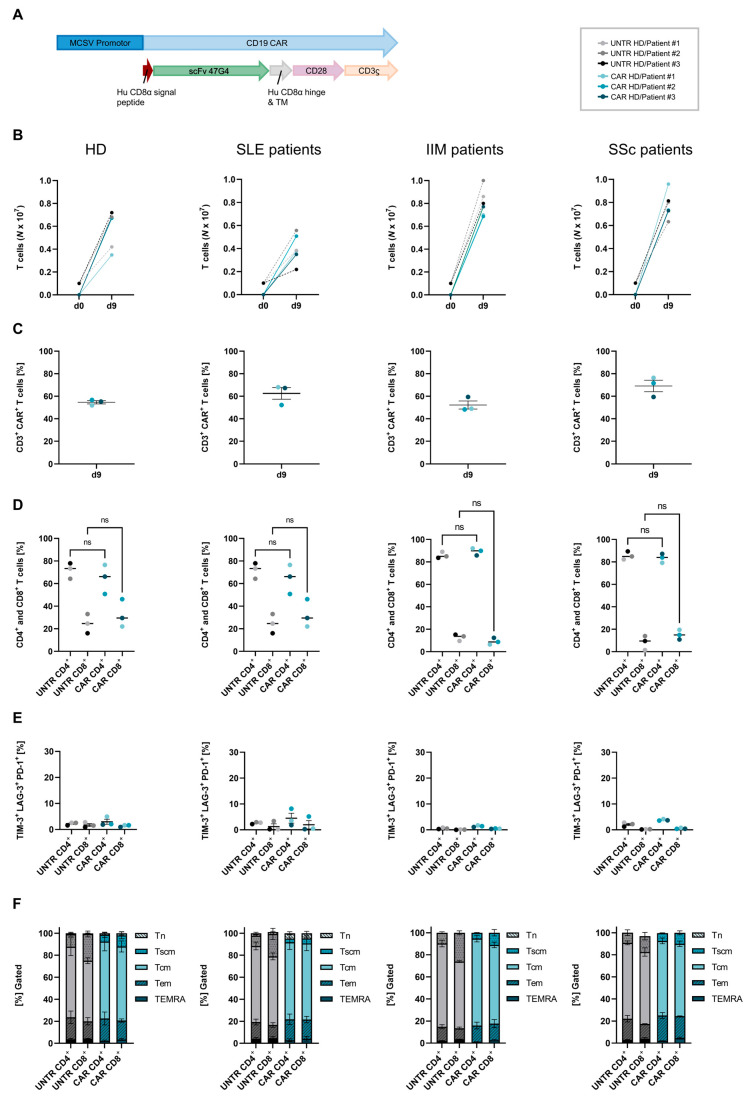
Characterization of CD19 CAR T-cells derived from SLE, IIM, and SSc patients and healthy donors. (**A**) Composition of the lentiviral CAR vector encoding a human (Hu) single-chain variable fragment (scFV) CD19-targeting domain, a CD8α hinge and transmembrane (TM) domain, a CD28 cytoplasmic costimulatory domain, and a CD3ζ cytoplasmic domain. (**B**) In vitro expansion per well of non-transduced (UNTR, dashed line) T-cells and transduced CAR T-cells (continuous line) derived from patients with SLE (N = 3), IIM (N = 3), or SSc (N = 3) and healthy donors (HDs; N = 3) between seeding on day 0 and harvest on day 9. (**C**) Transduction efficiency of SLE, IIM, SSc, and HD CAR T-cells assessed at day 9. (**D**) Percentage of CD4^+^ and CD8^+^ T-cells of SLE, IIM, and SSc patients and HDs within the CD3^+^ T-cell or CD3^+^ CAR^+^ T-cell population on day 9 after transduction. Data show individual values and were analyzed by Mann–Whitney test. Bars represent median, ns = not significant (**E**) Percentage of UNTR T-cells and CAR T-cells from SLE, IIM, and SSc patients and HDs co-expressing TIM-3^+^, LAG-3^+^, and PD-1 at day 9. Data show individual values with indication of mean. The data were analyzed by Tukey’s multiple comparison. (**F**) Distribution of memory phenotypes of CD4^+^ and CD8^+^ UNTR T-cells and CAR T-cells from SLE, IIM, and SSc patients and HDs. Naïve T-cells (Tn), stem cell memory T-cells (Tscm), central memory T-cells (Tcm), effector memory T-cells (Tem), and Tem re-expressing CD45RA (TEMRA) were determined by flow cytometry at day 9.

**Figure 3 ijms-26-00467-f003:**
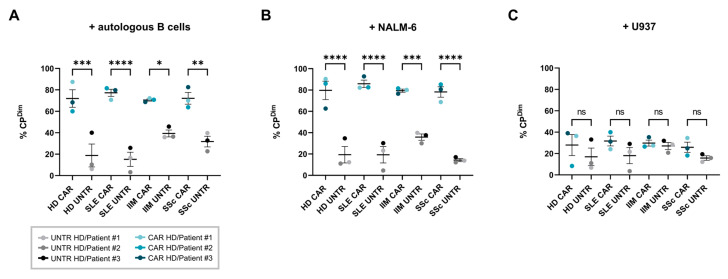
Proliferation of CAR T-cells from SLE, SSc, and IIM patients upon target cell engagement. Target T-cells were incubated with CAR T-cells or non-transduced (UNTR) T-cells derived from patients with SLE (N = 3), IIM (N = 3), and SSc (N = 3) and healthy donors (HDs; N = 3). Proliferation of CAR T-cells compared to UNTR T-cells incubated with (**A**) donor-matched autologous B-cells, (**B**) CD19^+^ B-cell line NALM-6 as positive control, and (**C**) CD19^−^ non-B-cell line U937 as negative control on day 4. Proliferating cells are determined by % Cytopainter CP^Dim^ gate, based on the UNTR T-cell monoculture as reference on day 4. Data show individual values with indication of mean. The data were analyzed by Tukey’s multiple comparison test, and significance is indicated by * *p* = 0.05, ** *p*= 0.01, *** *p* = 0.001 and **** *p* = 0.0001, ns = not significant.

**Figure 4 ijms-26-00467-f004:**
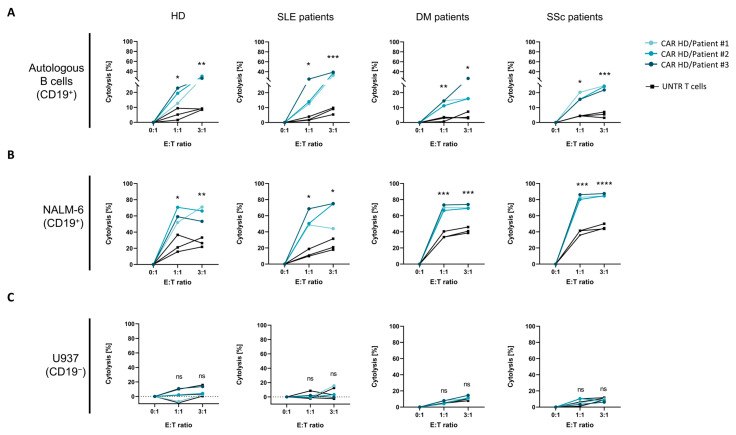
Cytolytic activity of CAR T-cells from SLE, IIM, and SSc patients and healthy donors. Target T-cells were co-cultured with CAR T-cells or non-transduced (UNTR) T-cells derived from patients with SLE (N = 3), IIM (N = 3), or SSc (N = 3) or healthy donors (HDs; N = 3) at E:T ratios of 0:1, 1:1, and 3:1. Cytolysis by patient-derived CAR T-cells (colored) and UNTR T-cells (black) incubated with (**A**) donor-matched autologous B-cells, (**B**) CD19^+^ B-cell line NALM-6 as positive control, or (**C**) CD19^−^ non-B-cell line U937 as negative control is defined as ((% live cells (target alone) − % live cells (sample of interest))/% live cells (target alone) × 100). The data were analyzed by Welch’s *t*-test, and significance is indicated by * *p* = 0.05, ** *p* = 0.01, *** *p* = 0.001, **** *p* = 0.0001, ns = not significant.

**Figure 5 ijms-26-00467-f005:**
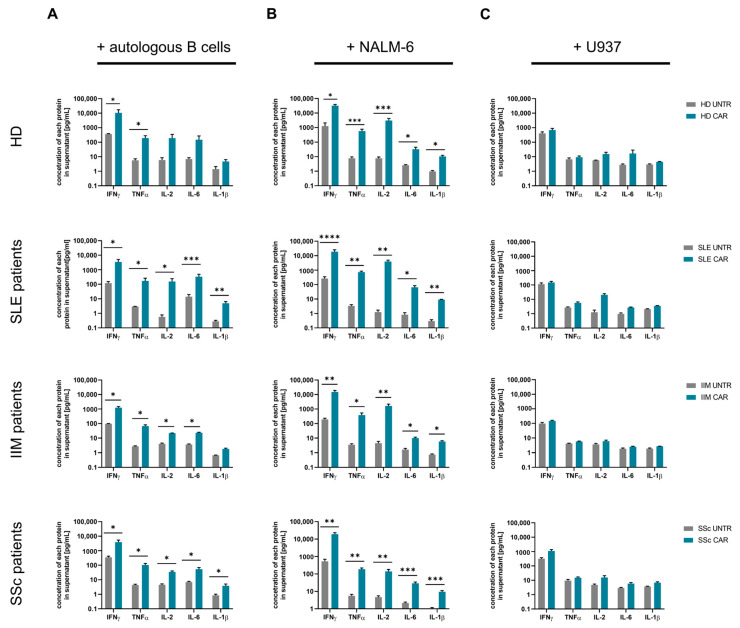
CAR T-cell mediated cytokine release after co-culture with target B-cells. Target T-cells were co-cultured with CAR T-cells or non-transduced (UNTR) T-cells derived from patients with SLE (N = 3), IIM (N = 3), and SSc (N = 3) or healthy donors (HDs; N = 3). Cytokine analysis of co-culture supernatants of SLE, IIM, SSc, and HD CAR T-cells compared to UNTR T-cells incubated with donor-matched (**A**) autologous B-cells, (**B**) CD19^+^ B-cell line NALM-6 as positive control, and (**C**) CD19^−^ non-B-cell line U937 as negative control. Values are shown as mean ± SEM. The data were analyzed by ratio paired *t*-test, * *p* = 0.05 and ** *p* = 0.01, *** *p* = 0.001, **** *p* = 0.0001.

**Table 1 ijms-26-00467-t001:** Patient characteristics.

Patient	Diagnosis	Age (Years)	Sex (F/M)	Disease Duration (Years)	Organ Involvement	Treatments (CT/PT)
Patients with systemic lupus erythematosus						
SLE P#1	Systemic lupus erythematosus	23	F	2	Skin, Joints, Lungs	CT: Hydroxychloroquine, Prednisolone, Anifrolumab
SLE P#2	Systemic lupus erythematosus	48	F	1	Skin, Heart, Kidney (class III)	CT: Hydroxychloroquine, MMF, Marcumar PT: Prednisolone, Belimumab, Deucravacitinib
SLE P#3	Systemic lupus erythematosus	39	F	9	Skin, Joints, HEM, Kidney (class IV)	CT: Quensyl, Azathioprine Prednisolone PT: Belimumab, Cyclophosphamide, Anifrolumab, Myfortic
Patients with systemic sclerosis						
SSc P#1	Systemic sclerosis	69	F	5	Skin, Raynaud, Esophagus, Lungs	CT: MMF
SSc P#2	Systemic sclerosis	68	F	3	Skin, Raynaud, Esophagus, Lungs	CT: MMF
SSc P#3	Systemic sclerosis	66	F	12	Skin, Joints, Lungs	CT: MMF, Nintedanib, Bosentan
Patients with idiopathic inflammatory myositis						
IIM P#1	Dermatomyositis (JO-1+)	48	F	6	Skin, Raynaud, Myositis, Fascitis, Lungs	CT: Filgotinib, MTX PT: MTX, Rituximab, Azathioprin, Prednisolone
IIM P#2	Dermatomyositis (JO-1+)	60	M	4	Skin, Raynaud, Esophagus, Lungs	CT: MMF PT: Prednisolone, MMF
IIM P#3	Dermatomyositis (JO-1+)	52	F	5	Skin, Raynaud, Joints, Lungs	CT: Nintedanib, Sildenafil, Hydroxychloroquine PT: MMF

F, female; M, male; CT, current therapy; PT, previous therapy; HEM, hematologic abnormalities; JO-1, antisynthetase syndrome; MMF, mycophenolate mofetil; MTX, methotrexate.

## Data Availability

All data are contained within the article.

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
