# Peer review of "Exploring CAR T-Cell Dynamics: Balancing Potent Cytotoxicity and Controlled Inflammation in CAR T-Cells Derived from Systemic Sclerosis and Myositis Patients"

_ijms, 2025, doi:10.3390/ijms26020467_

Round 1
Reviewer 1 Report
Comments and Suggestions for Authors
The authors take PBMCs from nine patients with three different autoimmune diseases, transduce them with anti-CD19 CARs, and measure their immunoreactivity to CD19+, autologous Bcells, or CD19- cell lines. Can the authors explain any novelty of this study, if the same CAR has been utilized previously for all these diseases to generate CARTs, and performed in patients?
Suggestions -
(1) Figures are very hard to interpret due to color schema/shading. If the authors could address this. Not every disease needs a separate color schema, but distinguishing betwixt groups is currently too difficult.
(2) FIG1B on graph show baseline amount of cells, or just list %yield, showing this graph is somewhat pointless. Again color chart is confusing. Alter so each column is the same color, but have different shapes/shading for Px 1-3
FIG2F - similarly, cannot determine Tnaive from Tcm.
(3) Is there a reason for separating FIG3 from FIG4? Isn't FIG3 is just one of the E:T ratios presented in FIG4?
Author Response
Comment: The authors take PBMCs from nine patients with three different autoimmune diseases, transduce them with anti-CD19 CARs, and measure their immunoreactivity to CD19+, autologous B cells, or CD19- cell lines. Can the authors explain any novelty of this study, if the same CAR has been utilized previously for all these diseases to generate CARTs, and performed in patients?
Response: Thank you for the comment. Our manuscript provides a detailed analysis of the manufacturing, functionality, and safety of Hu19-CD828Z CAR T cells in different autoimmune diseases patients and healthy donors. It emphasizes the translational potential of CAR T-cell therapy for systemic sclerosis, idiopathic inflammatory myositis, and systemic lupus erythematosus, and other autoimmune disorders. Overall, our manuscript provided novel findings on Hu19-CD828Z as potential fully human autologous CAR-T cell therapy, KYV-101, also for patients with SSc and IIM. The results are comparable to SLE settings in our manuscript. KYV-101 has been performed in patients with SLE patients, lupus nephritis, stiff person syndrome, myasthenia gravis, and multiple sclerosis. The IIM and SSc trails were performed with the autologous MB-CART19.1 CAR T-cell therapy (Miltenyi).
Suggestions of the reviewer-
Comment (1): Figures are very hard to interpret due to color schema/shading. If the authors could address this. Not every disease needs a separate color schema, but distinguishing betwixt groups is currently too difficult.
Response 1: Thank you for pointing this out. We changed all the style of all figures. Each column has the same color, but different shapes. Only CAR T cells are colored, untransduced T cells not (black). We hope, the figures can be interpreted more easily now.
Comment (2): FIG1B on graph show baseline amount of cells, or just list %yield, showing this graph is somewhat pointless. Again color chart is confusing. Alter so each column is the same color, but have different shapes/shading for Px 1-3
Response 2: Thank you for the comment. We changed the color setting (see above). Fig1B shows the viability of CD19+ B cells after enrichment. We think, that this point is important to show, because the viability of isolated B cells before adding them into the different experimental setups is essential for the outcome of the assays. We can show that the B cells from HD, but also from all autoimmune patients could be isolated in a comparable viable state.
Comment 2.1: FIG2F - similarly, cannot determine Tnaive from Tcm.
Response 2.1: We changed the shading of the different subsets in Figure 2F.
Comment (3): Is there a reason for separating FIG3 from FIG4? Isn't FIG3 is just one of the E:T ratios presented in FIG4?
Response 3: Thank you for your comment. We hope that we can explain this here. Figure 3 shows the proliferative activity of CAR-T cells from patients and HD upon contact to target cells with E:T ratio of 3:1 and focuses the T cells. Figure 4 demonstrates the cytolytic activity of CAR-T cells from SLE, IIM, SSc patients and HD with focus on the targeted B-cells. The % Cytolysis is defined as (% live cells (target alone)- % live cells (sample of interest))/ % live cells (target alone) x 100) and is therefore based on the living B cells.
Reviewer 2 Report
Comments and Suggestions for Authors
The manuscript provides a detailed analysis of the manufacturing, functionality, and safety of Hu19-CD828Z CAR T cells in autoimmune diseases patients and healthy donors. It emphasizes the translational potential of CAR T-cell therapy for systemic sclerosis (SSc), idiopathic inflammatory myositis (IIM), and systemic lupus erythematosus (SLE), and other autoimmune disorders. Overall, the manuscript provided novel findings on Hu19-CD828Z and I would recommend acceptance upon minor revisions.
1. The gating strategy and raw flow cytometry data are insufficiently detailed. Including representative gating figures in supplementary materials would be helpful.
2. Please provide more description of chimeric antigen receptor vector in Section 4, instead of just giving the references.
3. Please provide more mechanistic explanation on why CAR T-cells from HD showed a higher release of pro-inflammatory cytokines compared to CAR T-cells from autoimmune patients. How is it compared to allogenic CAT T cells or autologous CAR T cells from cancer patients?
4. Please proofread the manuscript for minor grammatical and typo issues and improve the quality/resolution of all the figures.
Author Response
The manuscript provides a detailed analysis of the manufacturing, functionality, and safety of Hu19-CD828Z CAR T cells in autoimmune diseases patients and healthy donors. It emphasizes the translational potential of CAR T-cell therapy for systemic sclerosis (SSc), idiopathic inflammatory myositis (IIM), and systemic lupus erythematosus (SLE), and other autoimmune disorders. Overall, the manuscript provided novel findings on Hu19-CD828Z and I would recommend acceptance upon minor revisions.
Comment 1. The gating strategy and raw flow cytometry data are insufficiently detailed. Including representative gating figures in supplementary materials would be helpful.
Response 1: Thank you for the comment. We added gating strategies for immunophenotyping (Suppl. Figure S1), Exhaustion and Memory phenotypes (Suppl. Figure S2, S3), Proliferation and Cytotoxicity (Suppl. Figure S4, S5). We refered to the suppl material on S. 12, 4.4; 4.6; 4.7 and S.5, 2.2.
Comment 2. Please provide more description of chimeric antigen receptor vector in Section 4, instead of just giving the references.
Response 2: Thank you for pointing this out. We added a more detailed description of vector design on S. 11, 4.2
Comment 3. Please provide more mechanistic explanation on why CAR T-cells from HD showed a higher release of pro-inflammatory cytokines compared to CAR T-cells from autoimmune patients. How is it compared to allogenic CAT T cells or autologous CAR T cells from cancer patients?
Response 3: Thank you for pointing this out. On S.10, part “Discussion” we say: “Hu19-CD828Z, the CAR vector used in KYV-101, is an anti-CD19 CAR construct with a CD28 co-stimulatory domain, engineered to release low cytokine levels by replacing the CD28 H/TM domain with a CD8α H/TM domain. This effect was previously demonstrated in vitro and was associated with reduced serum cytokine levels in B-cell lymphoma patients. As a result, decreased cytokine production correlated with a lower risk of CRS and reduced neurotoxicity in patients, without compromising clinical efficacy”. A deeper or direct comparison to allogenic CAR T cells or autologous CAR T cells from cancer patients is difficult and might be only descriptive. In our opinion the cytokine levels have to be measured in the same experiment for a direct comparison.
We added a more mechanistic explanation on why CAR T-cells from HD showed a higher release of pro-inflammatory cytokines compared to CAR T-cells from autoimmune patients: S. 11:” Interestingly, CAR T-cells from HD showed a higher release of INF-γ compared to CAR T-cells from autoimmune patients, suggesting possible modulation of inflammatory response due to the autoimmune disease background or prior therapies. Of note, some patients were on low-dose steroid treatment at the time of apheresis or get steroid treatments before, which could blunt the activity of the T-cell effector function. Despite this, the reduction efficacy in SLE patients appears to be maintained in vivo. Additionally, the lower cytokine release in response to autologous primary B-cells compared to NALM-6 B-cells suggests a milder cytokine release profile in autoimmune-targeting CAR T-cells, which may be beneficial in reducing the risk of CRS and ICANS in clinical applications.“ There might also be a dependency on the higher level of CD19 [MFI] on NALM-6 B-cells compared to primary B-cells.
Comment 4. Please proofread the manuscript for minor grammatical and typo issues and improve the quality/resolution of all the figures.
Response 4: Thank you for the comment. We checked for grammatical and typo issues and improved the quality/resolution of all the figures.